# Development and validation of a multivariable model to identify candidates for oral cancer screening in Nigeria
John Adeoye [1,2] ✉, Seidu A. Bello[3], Abdulwarith Akinshipo[4], Fadekemi O. Oginni[5], Bukola F. Adeyemi[6], Ramat O. Braimah[7], Ibrahim K. Suleiman[8], Mujtaba Bala[7], Hector O. Olasoji[8], Tosin Bakare[4], Taiwo Ajisebutu[3], Martina O. Mejabi[9], Emeka D. Odai[10], Ekosuehi T. Agho[11], Ifeoluwa Oketade[6], Victor I. Orji[12], Francis J. Bello[3], Nathan U. Ikimi[13], Deborah J. Enebong[13] & Yu-Xiong Su [1] ✉

## Abstract

**Background** Oral cancer screening can potentially improve the prevention and early detection of tumors if targeted toward at-risk individuals in the population. This study aims to profile the risk factors of oral cancer in a large Nigerian cohort to enable the selection of participants for cancer screening.

**Methods** This multicenter cross-sectional study involved an organized community oral cancer screening conducted among Nigerians between April 2023 and February 2024. Visual oral examination was conducted by trained personnel to determine the presence of oral cancer and precancerous lesions among participants. Additionally, we interviewed all screened participants based on thirty risk factor information items. Multivariate analysis was performed to determine factors that are significantly associated with oral cancer and precancerous conditions, which were used to construct a multivariate predictive model for oral cancer risk stratification.

**Results** Screening of 4049 participants detected 127 cases of oral cancer and precancerous lesions. Eight factors that are significantly associated with having a suspicious oral mucosal lesion at screening include tobacco smoking and snuff use, alcohol drinking, lack of fruits/vegetables consumption, and red/processed meat consumption, low spice consumption level, and comorbidities (p-value: <0.001–0.046). The predictive model based on the significant factors has an AUC of 0.74 (0.72 – 0.76) and Youden's index of 0.27 (0.25-0.29) that is higher than the metrics obtained for the conventional method of risk profiling for oral cancer (Youden's index: 0.25 (0.23-0.27)).

**Conclusions** Risk prediction model has better discrimination and net benefit than the conventional approach for identifying at-risk individuals for oral cancer. This finding supports the potential application of this method for risk stratification during targeted oral cancer screening.

## Plain language summary

Oral cancer screening is more effective when focused on people at high risk. Current programs only use the most common risk factors to decide who gets screened. This approach misses some high-risk individuals because it ignores other important risk factors. This study investigated key risk factors for oral cancer and precancerous conditions in a Nigerian population. The goal was to develop a risk prediction method for identifying people for screening. Our findings highlighted eight major risk factors. When combined, these factors slightly improved the identification of persons to be screened compared to the current approach. The method also offers more flexibility in determining how many people to screen, which could potentially save costs and resources, especially in developing countries.

Oral cancer is the most prevalent head and neck malignancy[1]. Globally, about 390,000 new cases and 190,000 deaths are recorded annually, and the majority of tumors are squamous cell carcinomas[1]. The 5-year survival rate of oral cancer has remained at about 50% for several decades due to the presentation of patients with advanced diseases[2–4]. Early detection of oral precancerous lesions and localized malignant tumors, achieved through

disease screening, can aid disease prevention and improve patient prognosis significantly[5].

Oral cancer is potentially amenable to disease screening and involves visual oral examination (VOE) and palpation by trained dental, medical, or frontline personnel[6,7]. Screening may be conducted using an organized or opportunistic approach. Due to the low prevalence of oral cancer among the

general population, mass screening is usually not recommended[8]. VOE screening to identify early oral cancer and precancerous lesions is beneficial when targeted toward at-risk individuals in the population[6,8,9]. Given that significant predisposing factors to developing oral cancer (including tobacco use, heavy alcohol consumption, and betel nut chewing) vary by geographical region, individualized risk profiling in different areas is crucial to the success of oral cancer screening. However, screening programs considering the high-risk approach have based risk stratification on the most prevalent factor in a region, which has the potential to miss out on other high-risk individuals without the most prevalent risk habits or from ethnic minorities who may practice less-prevalent risk habits[10-12].

Comprehensively determining the totality of risk factors and integrating them into a predictive model may improve the identification of at-risk individuals during oral cancer screening for better prevention of the disease[13,14]. This modeling approach to risk profiling has been investigated in some areas, such as the UK, Sri Lanka, India, and Hong Kong, and shown to perform better than the conventional unidimensional risk assessment method[13,15-19]. Compared to other regions, however, targeted oral cancer screening is infrequently performed in the African population, and little information is available on the totality of oral cancer risk in the region. Considering the limited access to care in many countries, knowing the risk profile can assist in streamlining awareness and promotion programs for better primary prevention of oral cancer. Furthermore, it would help in constructing sound risk prediction models to aid in patient selection for oral cancer screening and formulating public policy.

This study identifies eight risk factors that are significantly associated with the risk of oral cancer and precancerous lesions in the Nigerian population at screening using multiple logistic regression. They include tobacco smoking, tobacco snuff use, alcohol consumption, fruits, vegetables and red/processed meat consumption frequency, spice consumption level, and Charlson comorbidity index scores. Using these factors, we construct a multivariable risk prediction model to identify persons to be screened for oral cancer that is slightly better than the conventional method. This combined approach offers more flexibility in deciding the number of persons for screening, which is beneficial in resource-limited settings.

## Methods

### Study population
This multicenter cross-sectional study to assess risk factors for suspicious oral mucosal diseases and oral cancer is based on an organized community oral cancer screening program conducted at different centers within eight states/territory in Nigeria, i.e., Federal Capital Territory, Lagos, Borno, Sokoto, Osun, Oyo, Nasarawa, and Edo. These areas were selected out of the 36 states and the Federal Capital Territory as representatives of the six geopolitical zones in Nigeria. Lagos is the most populous state with a multiethnic population, while other states comprise populations that mostly belong to a particular ethnic group. Participants aged 30 years and above were invited to attend the screening programs in each state following community awareness and outreach exercises conducted between April 2023 and February 2024. Notably, participants with a previous history of oral cancer were excluded from analysis in this study.

### Oral cancer screening
Participants were screened for oral cancer and precancerous lesions by VOE and palpation. Specifically, the floor of the mouth and ventrolateral surface of the tongue were examined first in an anteroposterior direction bilaterally; thereafter, the dorsum, posterior lateral area and base of the tongue were examined. Lingual gingiva and retromolar area examinations were then performed, followed by inspection of the oropharynx. The hard palate and palatal gingiva were then checked for abnormalities afterward. Examination of the buccal and labial mucosa followed in a posteroanterior direction before concluding with inspection of the labial gingiva with the individual in centric occlusion. VOE for all participants was performed by forty-four calibrated dentists across all centers using wooden tongue depressors under white light. All examiners were trained to systematically examine various

sites of the oral mucosa by the lead author (JA) and were provided with ten images depicting different oral mucosa lesions to calibrate the reporting of suspicious oral mucosal conditions encountered following oral examination.

### Collection of risk factor information
Following VOE, an online interviewer-administered questionnaire (link) was used to collect information on various risk factors of oral cancer from all participants. Data were collected either directly into the electronic form (where the internet was available) or in hard copies of questionnaires and transferred to the electronic form subsequently. The questionnaires were initially designed based on a previous study in the Hong Kong population[8,10] and supplemented with additional information following a literature search to identify pertinent risk factors among the Nigerian population. The risk factor information collected included tobacco use history (smoking, chewing, and snuff use), alcohol consumption, second-hand tobacco smoking, betel nut chewing, kola nut chewing, diet history, family history of cancer, and comorbidity history. JA designed the questionnaire, while SB and AA provided face validity and qualitative content validity. The questionnaire was then pretested among 12 participants before deployment. Interviews were conducted by 125 calibrated personnel in this study. To standardize the interview framework across all states, all interviewers were trained by JA to use and interpret the questions before being tested using a model participant with their responses recorded.

### Outcomes and post-screening procedures
Outcome descriptions in this study were concordant with reports of oral cancer screening from other regions[10,16,20]. All participants were categorized as either 'screen-positive' or 'screen-negative' following VOE by the trained professionals. Positive status included the presence of early malignant lesions and potentially malignant disorders such as leukoplakia, erythroplakia, oral lichen planus, and reverse smokers' palate. Negative status meant that the participants had no oral mucosal conditions or that one or more benign mucosal conditions with no predisposition to oral cancer were observed. Examples are Fordyce's granules, leukoedema, frictional keratosis, aphthous stomatitis, geographic tongue, and hairy tongue. Participants with 'screen-positive' status were followed up by local clinicians for confirmatory diagnosis and management, while patients with negative status were educated on the risk factors and early signs of oral cancer and recommended for semiannual dental consultations for continuous oral cancer screening.

### Data analysis
Descriptive analysis was performed for all variables and presented in text, tables, and figures. Smoking pack years was calculated considering the different smoking devices as a product of the number of packs smoked per day and the duration of smoking. We also calculated the Charlson Comorbidity Index values using information on comorbidities provided by the participants. Binarization of variables such as educational level, tobacco smoking, and alcohol consumption was done to ensure an adequate event-per-variable value during analysis. We assessed the normal distribution of all continuous variables using Shapiro–Wilk's test before performing bivariate analysis to determine significant differences according to post-screening status (i.e., positive vs. negative) using the Mann–Whitney $U$ test. Furthermore, Pearson's Chi-Square test and Fisher's exact analysis were employed in comparing the differences in categorical variables. Multiple logistic regression was then performed, including all significant variables in bivariate analysis, in addition to tobacco smoking and alcohol consumption in line with previous studies[10,16].

### Predictive modeling
Significant risk factors in multivariate analysis were used as input features for a predictive algorithm to stratify oral cancer and precancerous lesions in the Nigerian population. This was conducted to ensure the predictive relevance of significant variables following multivariate analysis. We performed internal comparisons between different supervised learning

**Table 1 | Risk factor information of the cohort stratified by their screening status**

| Variables | | Screening status | | Total | p value |
|---|---|---|---|---|---|
| | | **Positive** (n = 127) | **Negative** (n = 3922) | **N (%)** | |
| Age (years) | Median (IQR) | 44 (33–53) | 43 (35–53) | 43 (35–53) | 0.858[a] |
| Sex | Female | 44 (34.6) | 1934 (49.3) | 1978 (48.9) | **0.001[b]** |
| | Male | 83 (65.4) | 1988 (50.7) | 2071 (51.1) | |
| Occupation | Artisan/Labor-related | 21 (16.5) | 791 (20.2) | 812 (20.1) | 0.254[b] |
| | Employed | 83 (65.4) | 2361 (60.2) | 2444 (60.4) | |
| | Professional | 5 (3.9) | 199 (5.1) | 204 (5.0) | |
| | Unemployed | 6 (4.7) | 322 (8.2) | 328 (8.1) | |
| | Others[d] | 12 (9.4) | 249 (6.3) | 261 (6.4) | |
| Educational status | Low | 89 (70.1) | 2834 (72.3) | 2923 (72.2) | 0.589[b] |
| | High | 38 (29.9) | 1088 (27.7) | 1126 (27.8) | |
| Tobacco smoking | Ever-smoker | 41 (32.3) | 494 (12.6) | 535 (13.2) | **1.13 × 10⁻¹⁰[b]** |
| | Non-smoker | 86 (67.7) | 3428 (87.4) | 3514 (86.8) | |
| Smoking pack years | Mean (SD) | 2.15 (6.56) | 0.51 (3.01) | 0.56 (3.18) | **1.13 × 10⁻¹⁰[a]** |
| Tobacco chewing | Ever-chewer | 5 (3.9) | 68 (1.7) | 73 (1.8) | 0.078[c] |
| | Non-chewer | 122 (96.1) | 3854 (98.3) | 3976 (98.2) | |
| Tobacco snuff use | Ever-user | 13 (10.2) | 96 (2.4) | 109 (2.7) | **0.0000277[c]** |
| | Non-user | 114 (89.8) | 3826 (97.6) | 3940 (97.3) | |
| Second-hand smoking exposure | Exposure | 47 (37.0) | 817 (20.8) | 864 (21.3) | **0.0000119[b]** |
| | No exposure | 80 (63.0) | 3105 (79.2) | 3185 (78.7) | |
| Alcohol consumption | Ever-drinker | 66 (52.0) | 1144 (29.2) | 1210 (29.9) | **3.31 × 10⁻⁸[b]** |
| | Non-drinker | 61 (48.0) | 2778 (70.8) | 2839 (70.1) | |
| Kola nut chewing | Ever-chewer | 50 (39.4) | 1190 (30.3) | 1240 (30.6) | **0.030[b]** |
| | Non-chewer | 77 (60.6) | 2732 (69.7) | 2809 (69.4) | |
| Tooth cleaning frequency | Daily | 118 (92.9) | 3595 (91.7) | 3713 (92.7) | 0.615[b] |
| | Not daily | 9 (7.1) | 327 (8.3) | 336 (8.3) | |
| Number of times per frequency (tooth cleaning) | Never | 0 | 12 (0.3) | 12 (0.3) | 0.609[b] |
| | Once | 90 (70.9) | 2891 (73.7) | 2981 (73.6) | |
| | Twice | 37 (29.1) | 1019 (26.0) | 1056 (26.1) | |
| Frequent gingival bleeding | Present | 34 (26.8) | 1150 (29.3) | 1184 (29.2) | 0.534[b] |
| | Absent | 93 (73.2) | 2772 (70.7) | 2865 (70.8) | |
| Removable denture use | Yes | 5 (3.9) | 102 (2.6) | 107 (2.6) | 0.388[c] |
| | No | 122 (96.1) | 3820 (97.4) | 3942 (97.4) | |
| Fruit consumption frequency | Doesn't consume | 7 (5.5) | 63 (1.6) | 70 (1.7) | **0.007[b]** |
| | Daily | 25 (19.7) | 888 (22.6) | 913 (22.5) | |
| | Alternate days | 13 (10.2) | 572 (14.6) | 585 (14.4) | |
| | Weekly | 26 (20.5) | 650 (16.6) | 676 (16.7) | |
| | Occasionally | 56 (44.1) | 1749 (44.6) | 1805 (44.6) | |
| Vegetables consumption frequency | Doesn't consume | 5 (3.9) | 52 (1.3) | 57 (1.4) | **0.024[b]** |
| | Daily | 27 (21.3) | 1082 (27.6) | 1109 (27.4) | |
| | Alternate days | 35 (27.6) | 860 (21.9) | 895 (22.1) | |
| | Weekly | 27 (21.3) | 711 (18.1) | 738 (18.2) | |
| | Occasionally | 33 (26.0) | 1217 (31.0) | 1250 (30.9) | |
| Fish/Seafood consumption frequency | Doesn't consume | 6 (4.7) | 91 (2.3) | 97 (2.4) | 0.114[b] |
| | Daily | 46 (36.2) | 1737 (44.3) | 1783 (44.0) | |
| | Alternate days | 23 (18.1) | 534 (13.6) | 557 (13.8) | |
| | Weekly | 19 (15.0) | 479 (12.2) | 498 (12.3) | |
| | Occasionally | 33 (26.0) | 1081 (27.6) | 1114 (27.5) | |
| Red/processed meat consumption frequency | Doesn't consume | 8 (6.3) | 327 (8.3) | 335 (8.3) | **0.031[b]** |
| | Daily | 35 (27.6) | 1234 (31.5) | 1269 (31.3) | |

**Table 1 (continued) | Risk factor information of the cohort stratified by their screening status**

| Variables | | Screening status | | Total | p value |
|---|---|---|---|---|---|
| | | **Positive** (n = 127) | **Negative** (n = 3922) | **N (%)** | |
| | Alternate days | 10 (7.9) | 469 (12.0) | 479 (11.8) | |
| | Weekly | 26 (20.5) | 463 (11.8) | 489 (12.1) | |
| | Occasionally | 48 (37.8) | 1429 (36.4) | 1477 (36.5) | |
| Consumption of spicy foods | Yes | 107 (84.3) | 3357 (85.6) | 3464 (85.6) | 0.672[b] |
| | No | 20 (15.7) | 565 (14.4) | 585 (14.4) | |
| Spiciness score | Median (IQR) | 4 (2–5) | 5 (3–6) | 5 (3 − 6) | **0.004[a]** |
| Consumption of hot beverages | Yes | 109 (85.8) | 3258 (83.1) | 3367 (83.2) | 0.414[b] |
| | No | 18 (14.2) | 664 (16.9) | 682 (16.8) | |
| Hotness score | Median (IQR) | 4 (3–6) | 5 (3–6) | 5 (3–6) | 0.081[a] |
| Frequency of Dental/ENT visits | Irregular | 111 (87.4) | 3608 (92.0) | 3719 (91.8) | 0.063[b] |
| | Regular | 16 (12.6) | 314 (8.0) | 330 (8.2) | |
| Use of alcohol containing mouthwash | Yes | 6 (4.7) | 77 (2.0) | 83 (2.0) | **0.045[c]** |
| | No | 121 (95.3) | 3845 (98.0) | 3966 (98.0) | |
| Number of family members with cancer | Median (Range) | 0 (0–3) | 0 (0–5) | 0 (0–5) | 0.172[a] |
| First degree relative with cancer | Yes | 5 (3.9) | 104 (2.7) | 109 (2.7) | 0.393[b] |
| Second/third degree relative with cancer | Yes | 3 (2.4) | 58 (1.5) | 61 (1.5) | 0.440[b] |
| Clinical type of familial cancer | Head and neck related | 2 (1.6) | 24 (0.6) | 26 (0.6) | 0.276[b] |
| | Not head and neck related | 6 (4.7) | 130 (3.3) | 136 (3.4) | |
| | Not applicable | 119 (93.7) | 3768 (96.1) | 3887 (96.0) | |
| Charlson comorbidity index | Mean (SD) | 0.30 (0.73) | 0.13 (0.38) | 0.14 (0.40) | **0.000426[a]** |

Values in bold are statistically significant, and all statistical tests were two-sided
[a]Mann–Whitney U test; [b]Pearson Chi-Square test; [c]Fisher's exact test
[d]Others include clergy, students, commercial sex workers, and military/security personnel

classifiers–random forest, decision trees, extremely randomized trees, adaptive boosting, and logistic regression (LR) leading to the selection of the LR classifier as the algorithm of choice. This study used a leave-one-site-out method to train and validate the model. In detail, we trained the LR model using data from seven states/territory (Federal Capital Territory, Borno, Sokoto, Osun, Oyo, Nasarawa, and Edo) before performing external testing using data from one state (Lagos). External testing was the basis for model selection in this study.

To address the imbalance in the screening status, we resampled the training data by generating synthetic samples using the Synthetic Minority Oversampling Technique with Edited Nearest Neighbors (SMOTE-ENN) and implemented balanced class weights within the model. SMOTE-ENN was used following its optimal performance compared to SMOTE and Adaptive Synthetic Sampling (ADASYN). Model training performance and stability were assessed using ten-fold cross-validation. To avoid data leakage, during training and cross-validation, we performed synthetic oversampling using nine out of ten data subsets, while the cross-validation accuracy was assessed using the single subset without imbalance correction. Label encoding and hyperparameter tuning were performed manually. The LR model was then evaluated using discrimination, calibration, and net benefit metrics. Discriminatory performance was assessed using the area under the receiver operating characteristic curve (AUC), sensitivity, specificity, and Youden's index, while Brier scores were used to assess the model's calibration. Also, Platt scaling was used to calibrate the predicted probabilities of the LR model. Decision curve analysis was performed at relevant threshold probabilities to determine the potential net benefit of the predictive model for risk stratification of oral cancer and precancerous lesions in Nigeria. Explainability of the model to determine the top factors that contribute most to high-risk or low-risk predictions using the test data was also implemented using Shapley's Additive Explanations (SHAP). Performance of the LR model (i.e., Youden's index and net benefit) was also compared with that of the current method for risk stratification in oral cancer screening, which

selects patients according to their tobacco smoking, alcohol consumption, and betel nut chewing status. In this method, persons who practiced any of these risk habits were deemed at-risk individuals and selected for screening, while those without the habits were classified as low-risk individuals.

### Statistics and reproducibility
Bivariate and multivariate data analyses were conducted using SPSS (version 28), while the risk prediction model was developed with Python (version 3.7) using the Scikit-learn machine learning library. During data analysis, we used information from 4049 screened participants, while during predictive modeling, data from 2748 participants were used for training/internal validation. Risk factor data for 1,301 participants were used for external testing. For all statistical tests, probability values below 5% indicated statistical significance.

### Ethics
Ethical approval to conduct this study was granted by the Federal Capital Territory Administration Medical Ethics Committee (reference number: FCTA/HHSS/HMB/ADH/130/23). Written and verbal informed consent was obtained from all study participants before enrollment in the study.

### Results
#### Descriptive characteristics and Risk factor analysis of screening cohort
A detailed description of the cohort and risk factor information collected is presented in Table 1. Overall, 4049 participants were screened for oral cancer and precancerous lesions from all study centers, with the majority of participants screened in Lagos (32.3%). Notably, 127 participants (3.1%) had a positive screening status for either oral cancer (0.5%) or precancerous conditions (2.7%), while 3922 (96.9%) participants screened negative for suspicious oral mucosal diseases. The median age (interquartile range: IQR) of all participants was 43 (35–53) year,s comprising slightly more males than

females (51.1 vs 48.9%). Upon stratifying the screening status by sex, the proportion of males that screened positive was significantly higher than those with a negative status (65.4% vs 50.7%; *p* = 0.001). Bivariate analyses further highlighted significant differences in tobacco smoking, smoking pack years, tobacco snuff use, second-hand smoking exposure, alcohol consumption, kola nut chewing, fruit, vegetable, and red meat consumption frequency, level of spice consumption, use of alcohol containing mouthwash, and Charlson comorbidity index between screen-negative and screen-positive participants (*p* < 0.001–0.045, Table 1).

### Multivariate analysis
Thirteen variables from bivariate analysis were included in a multiple logistic regression model. Findings are displayed in Table 2. Overall, eight variables, i.e., tobacco smoking, tobacco snuff use, alcohol consumption, fruits, vegetables and red/processed meat consumption frequency, spice consumption level, and Charlson comorbidity index scores, were significantly associated with suspicious oral mucosal diseases in the screening cohort. In detail, participants who smoked tobacco (either current or ex-smokers) were more likely to have a suspicious oral mucosal disease than non-smokers (OR: 2.01 (1.23–3.29); *p* = 0.006). Tobacco snuff use was also significantly associated with a positive screening status (OR: 2.46 (1.25–4.84); *p* = 0.009). Likewise, participants who consumed alcohol were more likely to have a positive screening status (OR: 1.98 (1.31–3.00); *p* = 0.001). Compared to those that did not consume fruits, participants who consumed fruits daily (OR: 0.38 (0.14–0.98)), on alternate days (OR: 0.24 (0.08–0.68)) and occasionally (OR: 0.37 (0.15–0.93)) were less likely to have a positive screening status (*p* = 0.008–0.046).

Vegetable consumption also appeared to be a protective factor for suspicious oral mucosal diseases in the cohort; however, a significant association was only observed among participants who consumed it daily (OR: 0.28 (0.10–0.82)) and occasionally (OR: 0.32 (0.11–0.93)) (*p* = 0.020–0.036). Red/processed meat consumption frequency was only significantly associated with a positive screening status among participants who consumed them weekly (OR: 2.37 (1.02–5.52); *p* = 0.045) compared to those who did not (*p* = 0.045). Multivariate analysis found an association between spice consumption level (graded on a 10-point scale) and screening status, with a 12% score reduction in the risk of suspicious oral mucosal diseases for a one-unit increase in the spice consumption level (*p* < 0.001). Also, for a one-unit rise in the Charlson comorbidity index scores, the risk of suspicious oral mucosal diseases at screening was observed to increase by 85% (*p* < 0.001).

### Predictive model performance, net benefit, and explainability
All eight independent associated factors for suspicious oral mucosal diseases in the screening cohort served as input features for the predictive model. We implemented the model by including the input variables sequentially; however, the spiciness score (ordinal) was uninformative and replaced by spice consumption (binary). Training data comprised information from 2748 participants (event rate: 3.2%), while data from 1301 participants (event rate: 3.1%) were used for external testing. The median (IQR) training AUC of the LR predictive model obtained following cross-validation was 0.65 (0.52–0.69) (Fig. 1a). Cross-validation AUC values for the comparator classifiers are also shown in Table S1.

Based on performance analysis using the external testing dataset, the AUC (95% CI) of the LR model was 0.74 (0.72–0.76), which was higher than the AUC values of random forest, extremely randomized trees, gradient boosting, and adaptive boosting (Table S1). The crude method had a sensitivity of 70% (67.5–72.5), specificity of 54.5% (51.8–57.2), and Youden's index of 0.25 (0.23–0.27). At the same specificity (54.5%), the LR model had a higher sensitivity of 72.5% (70.1–74.9) and Youden's index of 0.27 (0.25–0.29), suggesting better performance than the crude (current) method of identifying candidates for oral cancer screening. Regarding calibration, the Brier score (95% CI) of the LR model was 0.20 (0.18–0.22), which was lower than the values obtained for other classifiers, other than gradient boosting (Brier: 0.06 (0.05–0.07). Following Platt scaling of the LR model, the Brier score reduced to 0.03 (0.02–0.04). To confirm that the class

**Table 2 | Multiple logistic regression analysis of risk factors associated with suspicious oral mucosal diseases at screening**

| Variables | | Odds ratio (95% confidence interval) | *p* value[a] |
|---|---|---|---|
| Sex | Female | 1.00 | 0.356 |
| | Male | 1.22 (0.80–1.87) | |
| Tobacco smoking | Non-smoker | 1.00 | **0.006** |
| | Ever-smoker | 2.01 (1.23–3.29) | |
| Smoking pack years | | 1.03 (1.00–1.06) | 0.076 |
| Tobacco snuff use | Non-user | 1.00 | **0.009** |
| | Ever-user | 2.46 (1.25–4.84) | |
| Second-hand smoking exposure | No exposure | 1.00 | 0.078 |
| | Exposure | 1.47 (0.96–2.25) | |
| Alcohol consumption | Non-drinker | 1.00 | **0.001** |
| | Ever-drinker | 1.98 (1.31–3.00) | |
| Kola nut chewing | Non-chewer | 1.00 | 0.822 |
| | Ever-chewer | 1.05 (0.705–1.55) | |
| Fruit consumption frequency | Doesn't consume | 1.00 | |
| | Daily | 0.38 (0.14–0.98) | **0.046** |
| | Alternate days | 0.24 (0.08–0.68) | **0.008** |
| | Weekly | 0.41 (0.15–1.09) | 0.074 |
| | Occasionally | 0.37 (0.15–0.93) | **0.034** |
| Vegetable consumption frequency | Doesn't consume | 1.00 | |
| | Daily | 0.28 (0.10–0.82) | **0.020** |
| | Alternate days | 0.50 (0.17–1.48) | 0.199 |
| | Weekly | 0.40 (0.13–1.20) | 0.101 |
| | Occasionally | 0.32 (0.11–0.93) | **0.036** |
| Red/processed meat consumption frequency | Doesn't consume | 1.00 | |
| | Daily | 1.15 (0.51–2.59) | 0.732 |
| | Alternate days | 1.00 (0.38–2.67) | 1.000 |
| | Weekly | 2.37 (1.02–5.52) | **0.045** |
| | Occasionally | 1.60 (0.72–3.54) | 0.248 |
| Spiciness score | | 0.88 (0.82–0.94) | **0.000358** |
| Use of alcohol containing mouthwash | No | 1.00 | 0.119 |
| | Yes | 2.07 (0.83–5.17) | |
| Charlson comorbidity index scores | | 1.85 (1.38–2.49) | **0.0000395** |

[a]*p* values are based on multiple logistic regression analysis.
Values in bold are statistically significant, and all statistical tests were two-sided.

imbalance correction method was optimal, comparison with models trained with SMOTE (AUC: 0.38–0.58, Brier: 0.25–0.27) and ADASYN (AUC: 0.36–0.53, Brier: 0.25–0.27) showed that the latter techniques had poor discrimination and calibration (Table S2).

Upon stratifying the LR model's discriminative and calibration performance by demographic characteristics and risk habits, we found that the model had better AUC for predicting the need for screening among males (0.79) than females (0.62, Table S3). Also, a higher AUC was obtained for risk prediction among tobacco smokers, tobacco snuff users, and alcohol drinkers (Table S3). An explainability plot for the LR model using global SHAP values is shown in Fig. 2. Spice consumption, alcohol consumption, and tobacco smoking were the three most pertinent variables contributing to the predicted outputs in the testing dataset. Local explanation plots for specific cases are also presented in Fig. 3. Decision curve analysis showed

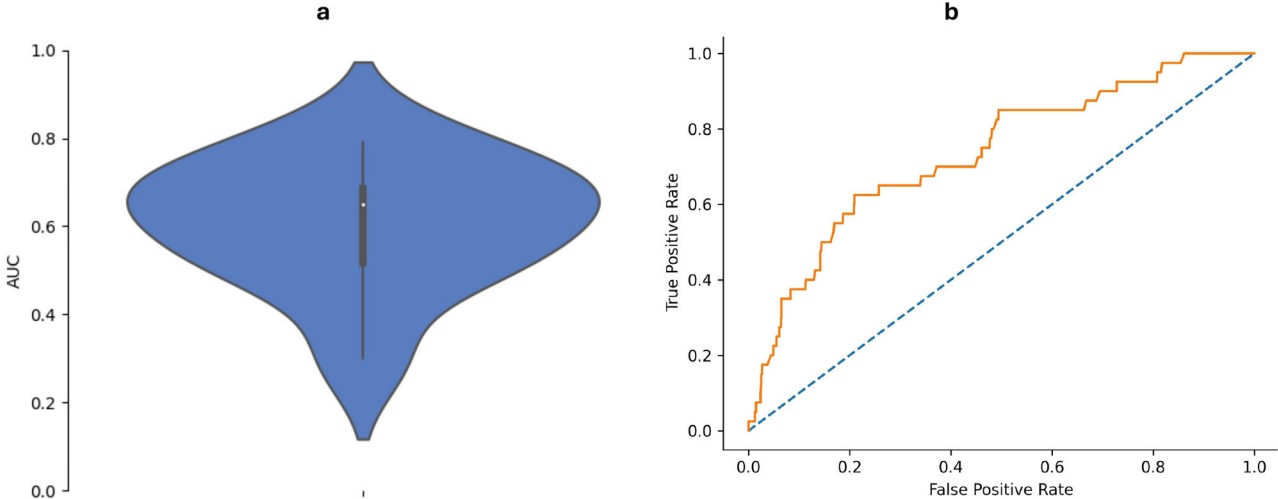

**Fig. 1 | Multivariable model performance. a** Violin plot of training AUC of the LR model following 10-fold cross-validation. **b** Receiver operating characteristic (ROC) curves for the LR model during external testing.

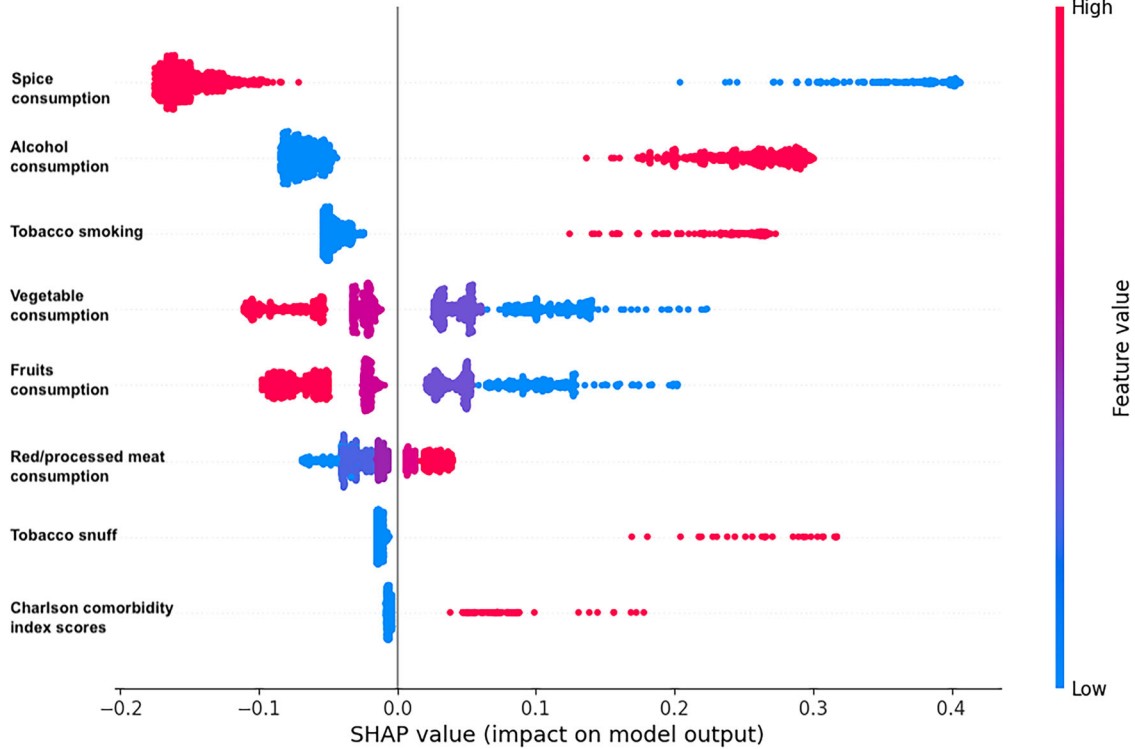

**Fig. 2 | Global explainability plot of the multivariable model.** SHAP summary plots of the predictive model showing the importance of the risk factors to predicted outputs in the external testing cohort.

that the LR model had a superior net benefit than the crude method in identifying participants with a likelihood of suspicious oral mucosal disease during screening (Fig. 4). Also, the decision curve for the LR model was higher than the decision curve obtained if all participants were screened for oral cancer and oral precancerous lesions.

## Discussion

Oral squamous cell carcinoma is the most common intraoral malignant tumor in Nigeria[21–23]. Since some reports have suggested that a large proportion of oral cancer patients in Nigeria are non-tobacco smokers and non-alcohol drinkers[24,25], this study aimed to identify pertinent risk factors to facilitate disease screening. Risk factor profiling of oral cancer and

precancerous lesions using a large screening cohort revealed eight factors that were significantly associated with these conditions in Nigeria. They include tobacco smoking, tobacco snuff use, alcohol drinking, lack of fruit and vegetable consumption, weekly red meat consumption, diets with less spice, and a high Charlson comorbidity index score. These significant variables were used to construct a multivariable model with an AUC and Brier score of 0.74 (0.72–0.76) and 0.20 (0.18–0.22) for predicting candidates at risk of suspicious oral mucosal diseases during oral cancer screening. Notably, the predictive model also had a higher Youden's index than the current (crude) method of risk stratification, with a higher potential net benefit if potentially employed in targeted oral cancer screening.

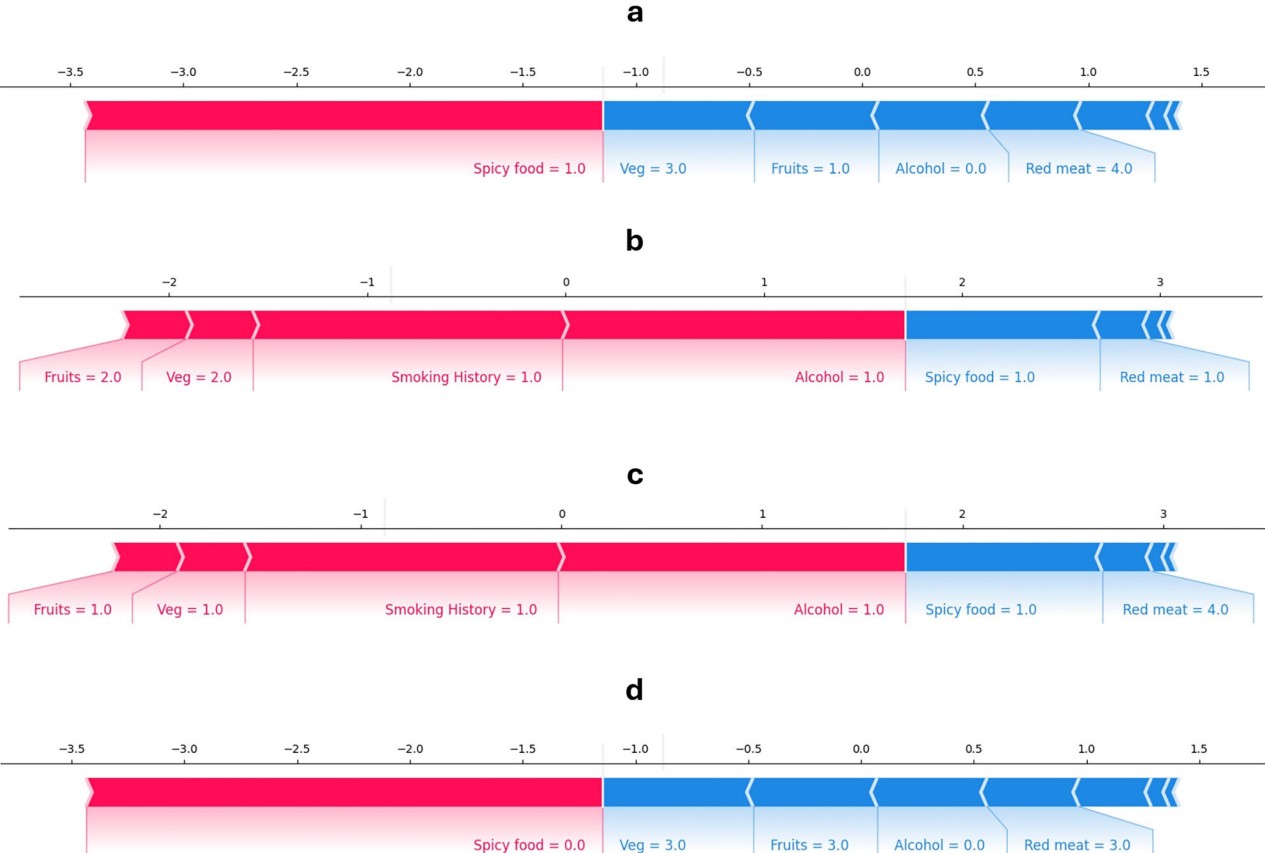

**Fig. 3 | Local explainability plots of the multivariable model. a** Force plot highlighting feature importance for a screen-negative case with a predicted probability of 0.10. **b** Force plot highlighting feature importance for a screen-negative case with a predicted probability of 0.96. **c** Force plot highlighting feature importance for a screen-positive case with a predicted probability of 0.92 **d** Force plot highlighting feature importance for a screen-positive case with a predicted probability of 0.10. (Keys - Spicy food (0 = No, 1-Yes), Vegetable intake frequency (Veg; 0 = Doesn't consume, 1 = Daily, 2 = Alternate days, 3 = Weekly, 4 = Occasionally), Fruits intake frequency (Fruits; 0 = Doesn't consume, 1 = Daily, 2 = Alternate days, 3 = Weekly, 4 = Occasionally), Red/processed meat intake (Red meat; 0 = Doesn't consume, 1 = Daily, 2 = Alternate days, 3 = Weekly, 4 = Occasionally), Smoking History (0=Non-smoker, 1-Ever-smoker), Alcohol consumption (Alcohol, 0=Non-drinker, 1-Ever-drinker)).

To our knowledge, this study is the first to comprehensively profile pertinent risk factors of oral cancer and oral precancerous conditions using a large screening cohort drawn from different geopolitical zones in Nigeria. The strength of this study includes the development of a risk prediction model based on these pertinent risk factors to select candidates for VOE in the Nigerian population, and better dichotomous discriminative ability of the model over the current method of risk stratification in targeted oral cancer screening. We also performed external testing by geographical region to assess model generalizability and evaluated the net benefit of the model to simulate its impact if deployed for real-world application.

Preliminary retrospective studies have previously highlighted some oral cancer risk factors in the Nigerian population. Adewole found alcohol drinking to be the most implicated factor in oral carcinogenesis among patients in Lagos[26]. Lawal et al also reported that low socioeconomic status, low fruit diet, and tobacco use increased oral cancer risk among patients in Ibadan[27]. In Eastern Nigerian patients, poor oral hygiene due to infrequent toothbrushing, chronic illnesses (comorbidities), malnutrition, and low socioeconomic status were found to be significantly associated with oral cancer development[25,28]. Likewise, in Maiduguri, oral cancer was significantly associated with poor diet and kola nut chewing[29].

Our findings confirmed the association of tobacco smoking, alcohol intake, low fruit or poor diet, and comorbidities with oral cancer and precancerous conditions in the Nigerian population. We also found that tobacco snuff was significantly associated with suspicious oral mucosal diseases in this study. However, this multicenter study did not corroborate the role of infrequent toothbrushing and kola nut chewing as predisposing factors for oral cancer and precancerous conditions in the population. The disparity in the findings may be due to the nature of our risk factor analysis, since multivariate analysis was performed to robustly select factors for oral cancer predisposition, as opposed to findings reported from bivariate analysis in earlier studies. However, this study did not investigate interactions between risk factors, which is an area for exploration future studies.

Previous studies among different populations have associated regular intake of red and/or processed meat with head and neck cancer[30,31]. Our study revealed that weekly consumption of red meat was an independent risk factor for oral cancer and precancerous lesions in the cohort. This may be related to red or processed meat containing/forming carcinogenic compounds such as polycyclic aromatic hydrocarbons and heterocyclic aromatic amines following intake[31]. While this reason is well investigated to explain the effect of red and processed meat in gastrointestinal tumors[32,33], studies confirming or elucidating new pathways specifically for head and neck cancers are needed. Notably, our findings showed that the level of spice consumption was significantly associated with oral cancer and precancerous conditions in Nigeria, suggesting that it is a protective factor. This observation was in line with our observations in the Hong Kong population[10] which showed that a low spice consumption level was associated with the risk of suspicious oral mucosal diseases at screening. Overall, this finding may be attributed to the potential chemopreventive action of phytochemicals such as capsaicin in chili peppers and curcumin in turmeric for oral squamous cell carcinoma. Nonetheless, more studies to understand the

**Fig. 4 | Net benefit of the multivariable model.**
Decision curve analysis plot of the LR model vs. crude method in stratifying patients for oral cancer screening in the external testing cohort.

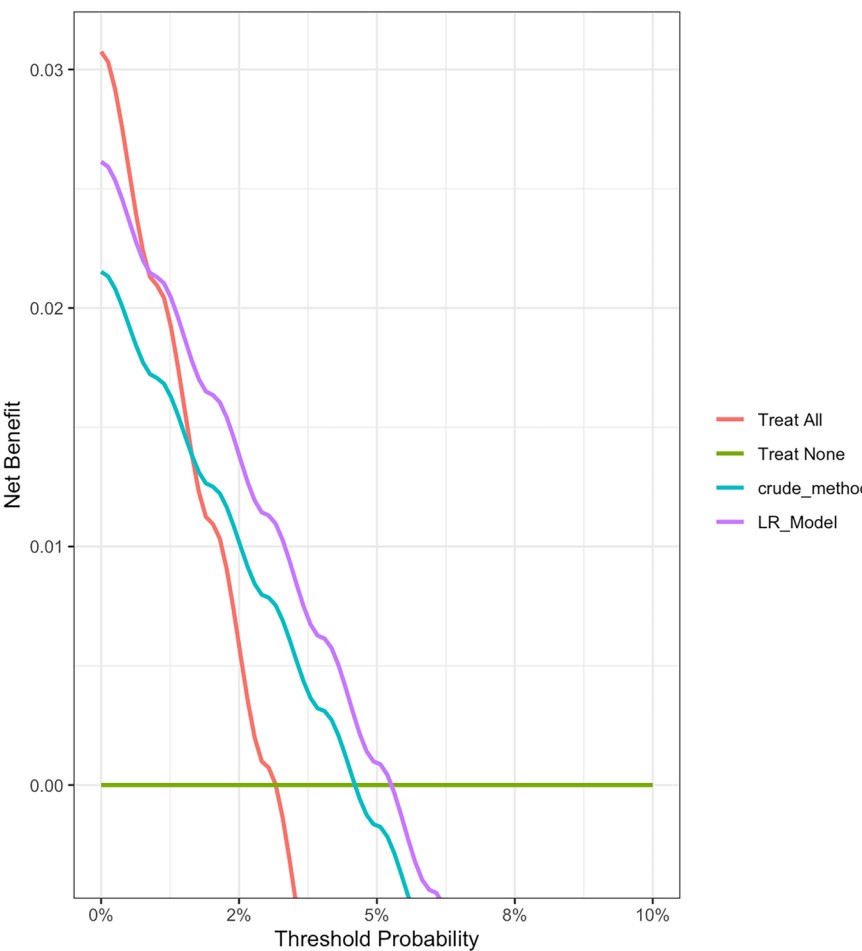

mechanism of action of phytochemicals in preventing oral carcinogenesis are imperative.

To showcase the predictive ability of the associated factors, we constructed a logistic regression model in this study. Considering that this approach had a better Youden's index and net benefit than conventional methods, our decision-support model can potentially assist targeted screening programs and frontline workers in the country to identify at-risk persons for oral cancer and precancerous conditions. Notably, the merit of the model is in its flexibility in selecting the number of patients for screening (especially in resource-limited settings) without sacrificing performance in risk profiling compared to the current/crude method of risk stratification in targeted oral cancer screening. Additionally, the predictive model developed in this study achieved similar discrimination ability and net benefit to those developed in the Hong Kong population[10]. However, further studies are essential to verify the effectiveness of the risk prediction model in Nigeria during oral cancer screening before deployment[34]. Notably, this study based performance comparison on the specificity of the crude method for selecting a threshold to determine the discriminative ability of the predictive model for effective comparison. We maintain that the threshold to stratify candidates for oral cancer screening using the predictive model should be based on the center of use/local validation, the goal of screening, and the region of application. As such, a dynamic threshold should be considered during the application of the model.

It is well known that oral cancer susceptibility exhibits great geographical variation; thus, the ability of this model to predict oral cancer susceptibility in other populations must be investigated. We, however, expect the model to perform satisfactorily among population groups with similar ethnicities, socioeconomic structures, and risk factor profiles as in Nigeria. Additionally, genetic predisposition to oral cancer was not assessed

in the cohort studied, and this could unravel unique biomarkers for risk stratification in the Nigerian population. This remains to be investigated in future studies. Future works involving the model constructed in this study should also aim to (i) collect and incorporate genomic information of oral cancer predisposition which could potentially improve the specificity, precision, and net benefit of the model (ii) consider developing an adaptive model and examine the effect of model updating (with prospective oral cancer screening data) on predictive performance, and (iii) perform cost-benefit analysis to determine the economic implications of applying the model to identify candidates for oral cancer screening compared to the traditional risk factor approach.

## Conclusions

Overall, this study identified tobacco smoking, tobacco snuff use, alcohol drinking, lack of fruit and vegetable consumption, weekly red meat consumption, diets with less spice, and a high Charlson comorbidity index score as significant risk factors for oral cancer and precancerous lesions in Nigeria. A logistic regression model based on these factors achieved an external testing AUC of 0.74 (0.72–0.76), Youden's index of 0.27 (0.25–0.29), and a Brier score of 0.20 (0.18–0.22) in selecting candidates for oral cancer screening. Furthermore, this risk prediction model had a higher net benefit than the current (crude) approach to identify at-risk individuals for screening, suggesting potential clinical and public health applications.

## Data availability

Data used in this study are not publicly available due to the need to maintain patient confidentiality, but anonymized spreadsheets may be available from the authors upon reasonable request, subject to meeting ethical and legal requirements for data sharing set by our institutions. For requests, please

contact John Adeoye via email at jadeoye@hku.hk. Source data for the figures in the article have been provided as Supplementary data.

## Code availability
Codes used to implement the multivariable model in this study are available online[35].

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

## Acknowledgements
The authors are grateful for the selfless contributions and commitment of the forty-four dentists and 125 calibrated personnel who participated in the screening and interviews of the study participants. We also appreciate all the study participants for their input. We thank the University of Hong Kong (HKU) Knowledge Exchange Strategic Impact Scheme (KE-SI-2022/23-19; J.A., S.A.B., A.A., Y-X.S.), Hong Kong Research Grants Council General Research Fund (17117523 and 17114722; J.A. and Y-X.S.), and HKU Seed Funding for Collaborative Research 2024/2025 (23017102377; J.A. and Y.-X.S.) for funding this work. All authors declare no conflicts of interest.

## Author contributions
J.A. and Y-X.S. were involved in study concepts, methodology, analysis, and writing of the original drafts of the manuscript. Y-XS provided supervision and sought funding for the work. S.A.B., A.A., F.O.O., B.F.A., R.O.B., I.K.S., M.B., H.O.O., T.B., T.A., M.O.M., E.D.O., E.T.A., I.O., V.I.O., F.J.B., N.U.I., and D.J.E. were involved in data collection and critically reviewed the manuscript. All authors approved the final version of the manuscript.

## Competing interests
The authors declare no competing interests.

## Additional information

[1]Faculty of Dentistry, Division of Oral and Maxillofacial Surgery, University of Hong Kong, Hong Kong, China. [2]Faculty of Dentistry, Division of Applied Oral Sciences and Community Dental Care, University of Hong Kong, Hong Kong, China. [3]Cleft and Facial Deformity Foundation, International Craniofacial Academy, Abuja, Nigeria. [4]Faculty of Dental Sciences, Department of Oral and Maxillofacial Pathology and Biology, University of Lagos, Lagos, Nigeria. [5]Faculty of Dentistry, Department of Oral and Maxillofacial Surgery, Obafemi Awolowo University, Ile-Ife, Nigeria. [6]Department of Oral Pathology, Faculty of Dentistry, University of Ibadan, Ibadan, Nigeria. [7]Faculty of Dental Sciences, Department of Oral and Maxillofacial Surgery, Usmanu Danfodiyo University, Sokoto, Nigeria. [8]Faculty of Dentistry, Department of Oral and Maxillofacial Surgery, University of Maiduguri, Borno, Nigeria. [9]Department of Dental Surgery, Federal Medical Center, Keffi, Nigeria. [10]Faculty of Dentistry, Department of Oral and Maxillofacial Surgery, University of Benin, Benin, Nigeria. [11]Department of Dental and Maxillofacial Surgery, National Hospital, Abuja, Nigeria. [12]Department of Family Dentistry, Federal Medical Centre, Keffi, Nigeria. [13]State House Medical Centre, Asokoro, Abuja, Nigeria. ✉e-mail: jaadeoye@connect.hku.hk; richsu@hku.hk

