## [Transparent Peer Review file · Communications Medicine]

DEVELOPMENT AND VALIDATION OF A MULTIVARIABLE MODEL TO IDENTIFY CANDIDATES FOR ORAL CANCER SCREENING IN NIGERIA

Corresponding Author: Dr John Adeoye

Version 0:

Reviewer comments:

Reviewer #1

(Remarks to the Author)

Using data from 4,049 participants in Nigeria screened for oral precancer and cancer, the authors developed and validated a logistic regression model to identify candidates for oral cancer screening. Overall, this was a fairly well-conducted analysis with some major weaknesses that are addressable.

Major weaknesses:

1. I think the term machine learning is overused and non-specific. I suggest changing the title to indicate a logistic regression model was used.

2. The authors used the Mann-Whitney U test (for continuous variables) and Chi-square/Fisher's exact test (for categorical variables) to detect variables with significant differences in post-screening status (positive vs. negative visual oral examination result). All significant variables were then included as predictors in a logistic regression (LR) model. A binary classifier was chosen to maximize sensitivity such that specificity >50%.

The event is rare with 127 events and 3,922 non-events in the dataset. Specificity is then nearly the same as the negative rate. So, does the last statement equate to dichotomizing by the median risk prediction? If so, I would state that for clarity.

3. The binary LR-based classifier is then compared to a dichotomous classifier of whether the individual was a risk habit user, using an ROC curve.

There is only one data point for each of the ROC curves. The lines drawn from that point to (0,0) and (1,1) are one of many possible interpolations of the data to a full ROC curve. It is possible to find possible ROC curves such that the binary LR-based classifier performs worse by AUC than the risk-habit-based classifier. The conclusion of the paper that risk prediction model has better discrimination than the conventional approach is therefore not supported by the analysis conducted.

4. To properly evaluate the LR-based classifier, I suggest drawing the full ROC curve based on evaluating a set of binary classifiers (e.g., risk from 0% to 100% at 1% increments). Then you can make an apples-to-apples comparison by assessing the sensitivity of the LR-based classifier versus the risk-habit-based classifier given the same specificity (and remove the statement in the results indicating that the risk-habit-based classifier have better specificity). I would drop mention of AUC for any binary classification; it is possible to report it for the continuous LR-based predicted risk. Afterwards, the predicted risk of the LR model can be dichotomized by the median of the risk distribution (as before) or at the same screening referral rate as the conventional approach or left to screening programs to define according to available screening resources/cost.

5. I do not understand what is represented in the cross-validation training accuracy and violin plot. Is this the training accuracy of each of the 10 validations, based off dichotomization using the median predicted risk? Please make the definition explicit and give a reference training accuracy to allow interpretation of whether performance is good or poor. For example, I believe that if I were to select randomly 95% of the population as negative, my training accuracy would be ~90%.

Other comments:

1. The results on descriptive characteristics, risk factor analysis, and multivariate analysis can be more concisely written. As currently presented, there is too much information described in text that is already shown in Table 1 and associations are presented twice, first as differences in distributions between screen-positive/negatives and then again as adjusted-HR from the logistic regression model. Of the two, the adjusted-HR is the preferred statistic.

Reviewed by:

Li C. Cheung

Principal Investigator

Biostatistics Branch, Division of Cancer Epidemiology and Genetics

National Cancer Institute, US NIH

Reviewer #2

(Remarks to the Author)

This manuscript is a well-executed study that addresses a pressing healthcare need with an innovative solution. While the methodology and results are strong, the addition of further clarifications, particularly around model performance, generalizability, and reproducibility, would significantly enhance its impact. By addressing these areas, the authors could elevate the work to a landmark study in the field of oral cancer screening.

Major Claims and Novelty

The manuscript introduces a machine learning (ML)-based logistic regression (LR) model designed for stratifying candidates for oral cancer screening in Nigeria. The following strengths underpin its novelty:

1. Integration of Localized Risk Factors: By including dietary patterns like spice consumption and region-specific habits such as tobacco snuff use, the study offers a culturally contextualized approach to oral cancer risk stratification.
2. Comparative Framework: Benchmarking the ML model against the current crude stratification method not only validates its added value but also addresses practical applications, which are often underexplored in AI-healthcare studies.
3. Explainability with SHAP: The use of explainability frameworks like SHAP provides transparency and aligns with best practices in ethical AI for healthcare.

While the work is undoubtedly novel, the following enhancements would further underscore its originality:

- Cross-referencing with Global Studies: Compare findings to other population-based ML studies (e.g., India, Sri Lanka) to contextualize how unique the Nigerian cohort's risk profile is.
- Discussion on Limitations of Previous Work: A brief critique of studies using unidimensional risk stratification would help justify the multidimensional modeling approach.

Validity and Evidence Strength

The authors have taken commendable steps to ensure the validity of their claims, including rigorous model training, validation, and multicenter data collection. However, the robustness of the conclusions could be enhanced by addressing the following areas:

1. Model Performance Metrics:

o The sensitivity (85%) of the LR model is notable, but the trade-off against specificity (51%) warrants further discussion. For example:

How would false positives affect the feasibility of implementing this model in screening programs with limited resources?
Could dynamic thresholds based on available resources or screening goals improve outcomes?

o The balanced accuracy (68%) is modest. Could ensemble methods (e.g., gradient boosting) outperform LR while maintaining explainability?

2. Risk Factor Analysis:

o Spice Consumption: While intriguing, the protective role of spices might have confounders (e.g., socioeconomic status or cultural differences in diet). Were confounders explicitly controlled in multivariate analysis?

o Red Meat Consumption: Weekly consumption emerged as a significant predictor, but the biological rationale (e.g., carcinogenic potential of processed meats) is not fully discussed. Adding references to carcinogenic pathways could strengthen this claim.

Statistical Analysis

The statistical methods are appropriately chosen but could benefit from further clarification and expansion:

1. Data Imbalance:

o While SMOTE-ENN was used to address class imbalance, the manuscript does not specify the performance impact of this resampling technique. Was there a comparison with simpler oversampling (e.g., SMOTE alone) or undersampling approaches? Including these results in supplementary material could bolster reproducibility.

2. Alternative Models:

o While LR is interpretable, its limitations in capturing nonlinear relationships are well-documented. Were models like random forests or gradient boosting compared in terms of performance and explainability? If excluded, a rationale should be provided.

o Could the authors explore hybrid models, where interpretable features are combined with black-box models, to enhance performance without compromising explainability?

3. Explainability:

o The SHAP summary plot is a strong addition, but global feature importance could be complemented by local explainability analyses. For instance:

Present specific patient cases where the model correctly or incorrectly classified risk.

Could SHAP force plots be included to visualize how risk factors combine to produce specific predictions?

4. Reproducibility Enhancements:

- o Providing pseudocode for the ML pipeline or access to the exact preprocessing steps (e.g., how categorical variables were encoded) would elevate reproducibility.
 - o Explicitly state whether all preprocessing steps were included within the cross-validation pipeline to avoid data leakage.
-

Future Directions

The authors briefly discuss future work, but more specific avenues could add depth:

1. Integration with Genomics:

o While genetic data is currently unavailable, future studies incorporating genomics could unravel additional biomarkers, improving model specificity.

2. Dynamic Model Updating:

o As more data becomes available, consider developing an adaptive model that updates itself with new information from screening programs.

3. Cost-Benefit Analysis:

o Quantify the economic implications of implementing the model versus traditional screening approaches to strengthen its case for real-world adoption.

Questions to Address

1. Could the authors elaborate on how misclassifications (false positives/negatives) were distributed among demographic or risk factor subgroups?

2. Were any interactions between risk factors (e.g., combined effects of smoking and alcohol) explored in the model? If not, this could be an area for future investigation.

Version 1:

Reviewer comments:

Reviewer #1

(Remarks to the Author)

Thank you for the revisions to the manuscript that addressed my prior comments.

In Figure 1b, the crude/conventional method should have just 1 data point on the AUC plot. The linear interpolation to (0,0) and (1,1) should be removed as they are interpolations, not real data. You should either remove the AUC for the crude/conventional method or present it as Youden's index but avoid directly comparing it to the AUC of the LR model.

At the specificity given by the crude/conventional method, the LR model appears to have the same sensitivity. This is probably to be expected as habit use is such a strong dichotomous predictor. As such, the LR model cannot be used to claim better performance than the crude/conventional method. However, you can claim that the LR model gives more flexibility in determining how many individuals to refer to screening, without sacrificing performance in risk profiling compared to the crude/conventional method.

This is the only comment I have but one that changes your conclusion.

Reviewers' comments:

Reviewer #1 (Remarks to the Author):

Using data from 4,049 participants in Nigeria screened for oral precancer and cancer, the authors developed and validated a logistic regression model to identify candidates for oral cancer screening. Overall, this was a fairly well-conducted analysis with some major weaknesses that are addressable.

Major weaknesses:

1. I think the term machine learning is overused and non-specific. I suggest changing the title to indicate a logistic regression model was used.

Response: Thank you for your comment. We have modified the title and the text to reflect your suggestion.

2. The authors used the Mann-Whitney U test (for continuous variables) and Chi-square/Fisher's exact test (for categorical variables) to detect variables with significant differences in post-screening status (positive vs. negative visual oral examination result). All significant variables were then included as predictors in a logistic regression (LR) model. A binary classifier was chosen to maximize sensitivity such that specificity > 50%. The event is rare with 127 events and 3,922 non-events in the dataset. Specificity is then nearly the same as the negative rate. So, does the last statement equate to dichotomizing by the median risk prediction? If so, I would state that for clarity.

Response: Thank you for the comment. Yes, we did that in the previous submission round. But after reading your comments and that of the other reviewer, we have now assessed the performance of the model using metrics that do not specify a threshold such as AUC and Brier scores alone. In the revised manuscript, we maintained that dynamic thresholds could be selected based on local center validation, goal of screening, and resource allocation.

3. The binary LR-based classifier is then compared to a dichotomous classifier of whether the individual was a risk habit user, using an ROC curve. There is only one data point for each of the ROC curves. The lines drawn from that point to (0,0) and (1,1) are one of many possible interpolations of the data to a full ROC curve. It is possible to find possible ROC

curves such that the binary LR-based classifier performs worse by AUC than the risk-habit-based classifier. The conclusion of the paper that risk prediction model has better discrimination than the conventional approach is therefore not supported by the analysis conducted.

Response: Thank you very much for this comment. You were right. We previously used the stratified risk status for ROC and AUC analysis. However, following your comments, we have used the predicted probabilities by the LR classifier to plot the ROC curve and calculate the model's AUC. From the curve, we see that at no point does the classifier perform worse than the conventional approach of risk stratification, which is in line with our conclusion. Please see the revised Figure 1 file.

4. To properly evaluate the LR-based classifier, I suggest drawing the full ROC curve based on evaluating a set of binary classifiers (e.g., risk from 0% to 100% at 1% increments). Then you can make an apples-to-apples comparison by assessing the sensitivity of the LR-based classifier versus the risk-habit-based classifier given the same specificity (and remove the statement in the results indicating that the risk-habit-based classifier have better specificity). I would drop mention of AUC for any binary classification; it is possible to report it for the continuous LR-based predicted risk. Afterwards, the predicted risk of the LR model can be dichotomized by the median of the risk distribution (as before) or at the same screening referral rate as the conventional approach or left to screening programs to define according to available screening resources/cost.

Response: Thank you for so much for your comment. We have now reported AUC based on the predicted probabilities instead of binary classification. At all specificities, the LR model appeared to outperform the conventional approach.

5. I do not understand what is represented in the cross-validation training accuracy and violin plot. Is this the training accuracy of each of the 10 validations, based off dichotomization using the median predicted risk? Please make the definition explicit and give a reference training accuracy to allow interpretation of whether performance is good or poor. For example, I believe that if I were to select randomly 95% of the population as negative, my training accuracy would be ~90%.

Response: Thank you so much for your comment. We have now discarded the accuracy metrics and revised the performance analysis and violin plot by performing and reporting the AUC of the model which we calculated with the predicted probabilities for the test data in the cross-validation folds.

Other comments:

1. The results on descriptive characteristics, risk factor analysis, and multivariate analysis can be more concisely written. As currently presented, there is too much information described in text that is already shown in Table 1 and associations are presented twice, first as differences in distributions between screen-positive/negatives and then again as adjusted-HR from the logistic regression model. Of the two, the adjusted-HR is the preferred statistic.

Response: Thank you for your comment. We have revised the bivariate analysis to be more concise and underscore the adjusted odds ratio as the preferred statistic for associations.

Reviewer #2 (Remarks to the Author):

This manuscript is a well-executed study that addresses a pressing healthcare need with an innovative solution. While the methodology and results are strong, the addition of further clarifications, particularly around model performance, generalizability, and reproducibility, would significantly enhance its impact. By addressing these areas, the authors could elevate the work to a landmark study in the field of oral cancer screening.

Major Claims and Novelty

The manuscript introduces a machine learning (ML)-based logistic regression (LR) model designed for stratifying candidates for oral cancer screening in Nigeria. The following strengths underpin its novelty:

1. Integration of Localized Risk Factors: By including dietary patterns like spice consumption and region-specific habits such as tobacco snuff use, the study offers a culturally contextualized approach to oral cancer risk stratification.
2. Comparative Framework: Benchmarking the ML model against the current crude stratification method not only validates its added value but also addresses practical applications, which are often underexplored in AI-healthcare studies.
3. Explainability with SHAP: The use of explainability frameworks like SHAP provides transparency and aligns with best practices in ethical AI for healthcare.

While the work is undoubtedly novel, the following enhancements would further underscore

its originality:

- Cross-referencing with Global Studies: Compare findings to other population-based ML studies (e.g., India, Sri Lanka) to contextualize how unique the Nigerian cohort's risk profile is.
- Discussion on Limitations of Previous Work: A brief critique of studies using unidimensional risk stratification would help justify the multidimensional modeling approach.

Response: Thank you for highlighting all these!

Validity and Evidence Strength

The authors have taken commendable steps to ensure the validity of their claims, including rigorous model training, validation, and multicenter data collection. However, the robustness of the conclusions could be enhanced by addressing the following areas:

1. Model Performance Metrics:

- o The sensitivity (85%) of the LR model is notable, but the trade-off against specificity (51%) warrants further discussion. For example:
 - ♣ How would false positives affect the feasibility of implementing this model in screening programs with limited resources?
 - ♣ Could dynamic thresholds based on available resources or screening goals improve outcomes?

Response: Thank you for these comments! Based on your comments, we have refrained from constraining the specificity and left the discriminative performance assessments to using AUC instead of selecting a single threshold for risk stratification. This way, future application can select threshold based on local validation performance, screening goals, and resources available.

- o The balanced accuracy (68%) is modest. Could ensemble methods (e.g., gradient boosting) outperform LR while maintaining explainability?

Response: Thank you for this comment. We previously performed comparisons with ensemble classifiers, tabular deep learning methods, and LR before model selection but did not report it. In the revised manuscript, we have now updated the results to include this data. Our results showed that LR had better discriminative performance than ensemble methods.

2. Risk Factor Analysis:

o Spice Consumption: While intriguing, the protective role of spices might have confounders (e.g., socioeconomic status or cultural differences in diet). Were confounders explicitly controlled in multivariate analysis?

Response: Thank you for your comment. Significant confounders were controlled in the multivariable model. Actually, we performed similar analysis in the Hong Kong population among different ethnicities and spice consumption also appeared to have a protective role in that population.

o Red Meat Consumption: Weekly consumption emerged as a significant predictor, but the biological rationale (e.g., carcinogenic potential of processed meats) is not fully discussed. Adding references to carcinogenic pathways could strengthen this claim.

Response: Thank you for your comment. We have revised the discussion section to include some sentences providing possible explanations on the carcinogenic potential of red and processed meat.

Statistical Analysis

The statistical methods are appropriately chosen but could benefit from further clarification and expansion:

1. Data Imbalance:

o While SMOTE-ENN was used to address class imbalance, the manuscript does not specify the performance impact of this resampling technique. Was there a comparison with simpler oversampling (e.g., SMOTE alone) or undersampling approaches? Including these results in supplementary material could bolster reproducibility.

Response: Thank you for your comment. We have presented results comparing SMOTE-ENN to SMOTE alone and ADASYN. The latter techniques performed significantly poorer than SMOTE-ENN and we retained it for model development.

2. Alternative Models:

o While LR is interpretable, its limitations in capturing nonlinear relationships are well-documented. Were models like random forests or gradient boosting compared in terms of performance and explainability? If excluded, a rationale should be provided.

Response: Thank you for comment! We performed comparisons to tree-based techniques and the LR model performed better. Results are in Table S1 (supplementary file) and reported briefly in the manuscript text.

o Could the authors explore hybrid models, where interpretable features are combined with black-box models, to enhance performance without compromising explainability?

Response: Thank you for your comment. We also explored models like MLP and TabNet but they did not outperform LR. So, we retained this classifier in the study.

3. Explainability:

o The SHAP summary plot is a strong addition, but global feature importance could be complemented by local explainability analyses. For instance:

- ♣ Present specific patient cases where the model correctly or incorrectly classified risk.
- ♣ Could SHAP force plots be included to visualize how risk factors combine to produce specific predictions?

Response: We really appreciate this comment! We have now included local explanations to buttress the global SHAP explanations. Force plots for four predicted outputs are included as Figure 3.

4. Reproducibility Enhancements:

- o Providing pseudocode for the ML pipeline or access to the exact preprocessing steps (e.g., how categorical variables were encoded) would elevate reproducibility.
- o Explicitly state whether all preprocessing steps were included within the cross-validation pipeline to avoid data leakage.

Response: Thank you for your comment. To ensure reproducibility, we have deposited the codes in GitHub and added a link in the manuscript for readers to access and implement if needed

Future Directions

The authors briefly discuss future work, but more specific avenues could add depth:

1. Integration with Genomics:

- o While genetic data is currently unavailable, future studies incorporating genomics could

unravel additional biomarkers, improving model specificity.

2. Dynamic Model Updating:

o As more data becomes available, consider developing an adaptive model that updates itself with new information from screening programs.

3. Cost-Benefit Analysis:

o Quantify the economic implications of implementing the model versus traditional screening approaches to strengthen its case for real-world adoption.

Response: Thank you for your comments! We have included the points mentioned in the discussion and expanded on the study limitations and future directions.

Questions to Address

1. Could the authors elaborate on how misclassifications (false positives/negatives) were distributed among demographic or risk factor subgroups?

Response: Thank you so much for this comment. We have now stratified the AUC and Brier scores by demographic variables and risk habits. Results are shown in Table S3 and reported in the text.

2. Were any interactions between risk factors (e.g., combined effects of smoking and alcohol) explored in the model? If not, this could be an area for future investigation.

Response: Thank you for your comment. We have included this as an area for future investigation in the manuscript.

Reviewers' comments:

Reviewer #1 (Remarks to the Author):

Thank you for the revisions to the manuscript that addressed my prior comments.

In Figure 1b, the crude/conventional method should have just 1 data point on the AUC plot. The linear interpolation to (0,0) and (1,1) should be removed as they are interpolations, not real data. You should either remove the AUC for the crude/conventional method or present it as Youden's index but avoid directly comparing it to the AUC of the LR model.

Response: Thank you for your comments. We have removed the AUC curve for the conventional method in Figure 1b and used the Youden's index for comparison.

At the specificity given by the crude/conventional method, the LR model appears to have the same sensitivity. This is probably to be expected as habit use is such a strong dichotomous predictor. As such, the LR model cannot be used to claim better performance than the crude/conventional method. However, you can claim that the LR model gives more flexibility in determining how many individuals to refer to screening, without sacrificing performance in risk profiling compared to the crude/conventional method.

This is the only comment I have but one that changes your conclusion.

Response: Thank you so much for this comment. Please note that at the same specificity as the conventional method (54.5%), the LR model had a sensitivity of 72.5% compared to the 70% sensitivity of the conventional method. As such, the Youden's index of the LR model is higher (0.27 vs 0.25), this supports the conclusion of the study still. To further strengthen our conclusions and discussions, we have also highlighted the use of the model to determine how many persons are referred for screening as a crucial merit that supports application.